# DIFFUSION MODELS FOR TABULAR DATA IMPUTATION AND SYNTHETIC DATA GENERATION

## ABSTRACT

Data imputation and data generation are crucial tasks in various domains, ranging from healthcare to finance, where incomplete or missing data can hinder accurate analysis and decision-making. In this paper, we explore the use of diffusion models with transformer conditioning for both data imputation and data generation tasks. Diffusion models have recently emerged as powerful generative models capable of capturing complex data distributions. By incorporating transformer conditioning, we harness the ability of transformers to model dependencies and cross-feature interactions within tabular data. We conduct a comprehensive evaluation by comparing the performance of diffusion models with transformer conditioning against state of the art techniques such as Variational Autoencoders (VAEs) and Generative Adversarial Networks (GANs) on benchmark datasets. The benchmark focuses on the assessment of generated samples with respect to Machine Learning (ML) utility, statistical similarity, and privacy risk. For the task of data imputation, our evaluation centers on the utility of the generated samples across different levels of missing features.

## 1 INTRODUCTION

Recent advancements in deep generative models, particularly Generative Adversarial Networks (GANs) (Goodfellow et al., 2014) and Diffusion Models (Sohl-Dickstein et al., 2015; Ho et al., 2020), now provide the ability to both model complex probability distributions and draw high-quality samples. Such capabilities have found applications in domains like image and audio processing (Zhang et al., 2022; Rombach et al., 2022; Liu et al., 2023) and have extended their utility to tabular data generation (Xu et al., 2019; Kotelnikov et al., 2023).

Particularly for tabular data, synthetic data stands out as a privacy-preserving alternative to real data. It enables the generation of datasets that emulate the statistical properties of their original counterparts, all while reducing the risk of individual privacy leakage. This generation of new samples can provide value by improving existing databases, like for example rectifying class imbalances, reducing biases, or expanding the size of smaller datasets. Moreover, integrating methods for differential privacy (Dwork, 2006; Jälkö et al., 2021) with generative models for tabular data makes it possible for organizations to distribute synthetic user data amongst their teams without legal concerns.

In this study, we consider synthetic data generation as a general case of data imputation. In instances where every column in a sample has missing values, the task of data imputation naturally transitions to synthesizing new data. We introduce *TabGenDDPM*, a new conditioning in diffusion model for tabular data using a transformer and special masking mechanism that makes it possible to tackle both tasks with a single model.

The key contributions of this work include:

- incorporation of a transformer within the diffusion model to model inter-feature interactions better within tabular data.
- an innovative masking and conditioning strategy on features, enabling both data imputation and generation with a single model.
- our approach outperforms state-of-the-art baselines in evaluation of the generated data regarding Machine Learning (ML) utility and statistical similarity with comparable privacy risk.

## 2 RELATED WORK

**Diffusion Models** First introduced by Sohl-Dickstein et al. (2015); Ho et al. (2020), diffusion models utilize a two-step generative approach. Initially, they degrade a data distribution using a forward diffusion process by continuously introducing noise from a known distribution. Following this, they employ a reverse process to restore the data's original structure. At their core, these models leverage parameterized Markov chains, starting typically from a foundational distribution such as a standard Gaussian, and use deep neural networks to reverse the diffusion. As evidenced by recent advancements (Nichol & Dhariwal, 2021; Dhariwal & Nichol, 2021), diffusion models have showcased their capability, potentially surpassing GANs in image generation capabilities. The specialized adaptation of this approach, *TabDDPM* (Kotelnikov et al., 2023), focuses on tabular data generation.

**Data Imputation** Handling missing values in datasets is a significant challenge. Traditional approaches might involve removing rows or columns with missing entries or filling gaps with average values for a particular feature. However, recent trends are shifting towards ML techniques (Van Buuren & Groothuis-Oudshoorn, 2011; Bertsimas et al., 2017) and deep generative models (Yoon et al., 2018; Biessmann et al., 2019; Wang et al., 2021b; Ipsen et al., 2022) for this purpose.

**Generative models** for tabular data are getting more attention in ML (Xu et al., 2019; Engelmann & Lessmann, 2021; Jordon et al., 2018; Fan et al., 2020; Torfi et al., 2022; Zhao et al., 2021; Kim et al., 2021; Zhang et al., 2021; Nock & Guillame-Bert, 2022; Wen et al., 2022). Notably, tabular VAEs (Xu et al., 2019) and GANs (Xu et al., 2019; Engelmann & Lessmann, 2021; Jordon et al., 2018; Fan et al., 2020; Torfi et al., 2022; Zhao et al., 2021; Kim et al., 2021; Zhang et al., 2021; Nock & Guillame-Bert, 2022; Wen et al., 2022) have shown promise in this field. Recently, *TabDDPM* has emerged as a powerful method for tabular data generation, leveraging the strengths of Diffusion Models. Our research builds upon *TabDDPM*, targeting both tabular data generation and imputation.

## 3 BACKGROUND

**Diffusion models**, as introduced by (Sohl-Dickstein et al., 2015; Ho et al., 2020), involve a two-step process: first degrading a data distribution using a forward diffusion process and then restoring its structure through a reverse process. Drawing insights from non-equilibrium statistical physics, these models employ a forward Markov process which converts a complex unknown data distribution into a simple known distribution (e.g., a Gaussian) and vice-versa a generative reverse Markov process that gradually transforms a simple known distribution into a complex data distribution.

More formally, the forward Markov process $q\left(x_{1:T}|x_0\right) = \prod_{t=1}^{T} q\left(x_t|x_{t-1}\right)$ gradually adds noise to an initial sample $x_0$ from the data distribution $q\left(x_0\right)$ sampling noise from the predefined distributions $q\left(x_t|x_{t-1}\right)$ with variances $\{\beta_1, \ldots, \beta_T\}$. Here $t \in [1, T]$ is the timestep, $T$ is the total number of timesteps used in the forward/reverse diffusion processes and $1 : T$ means the range of timesteps from $t = 1$ to $t = T$.

The reverse diffusion process $p\left(x_{0:T}\right) = \prod_{t=1}^{T} p\left(x_{t-1}|x_t\right)$ gradually denoises a latent variable $x_T \sim q\left(x_T\right)$ and allows generating new synthetic data. Distributions $p\left(x_{t-1}|x_t\right)$ are approximated by a neural network with parameters $\theta$. The parameters are learned optimizing a variational lower bound (VLB):

$$L_{\text{vlb}} := L_0 + L_1 + \ldots + L_{T-1} + L_T \tag{1}$$

$$L_0 := -\log p_\theta(x_0|x_1) \tag{2}$$

$$L_{t-1} := D_{KL}(q(x_{t-1}|x_t, x_0) \,||\, p_\theta(x_{t-1}|x_t)) \tag{3}$$

$$L_T := D_{KL}(q(x_T|x_0) \,||\, p(x_T)) \tag{4}$$

The term $q\left(x_{t-1}|x_t, x_0\right)$ is the *forward process posterior distribution* conditioned on $x_t$ and on the initial sample $x_0$. $L_{t-1}$ is the Kullback-Leibler divergence between the posterior of the forward process and the parameterized reverse diffusion process $p_\theta\left(x_{t-1}|x_t\right)$.

**Gaussian diffusion models** operate in continuous spaces $(x_t \in \mathbb{R}^n)$ and in this case the aim of the forward Markov process is to convert the complex unknown data distribution into a known Gaussian

distribution. This is achieved by defining a forward noising process $q$ that given a data distribution $x_0 \sim q(x_0)$, produces latents $x_1$ through $x_T$ by adding Gaussian noise at time $t$ with variance $\beta_t \in (0, 1)$.

$$q\left(x_t|x_{t-1}\right) := \mathcal{N}\left(x_t; \sqrt{1 - \beta_t}x_{t-1}, \beta_t I\right)$$

$$q\left(x_T\right) := \mathcal{N}\left(x_T; 0, I\right)$$

(5)

If we know the exact reverse distribution $q(x_{t-1}|x_t)$, by sampling from $x_T \sim \mathcal{N}(0, I)$, we can execute the process backward to obtain a sample from $q(x_0)$. However, given that $q(x_{t-1}|x_t)$ is influenced by the complete data distribution, we employ a neural network for its estimation:

$$p_\theta\left(x_{t-1}|x_t\right) := \mathcal{N}\left(x_{t-1}; \mu_\theta\left(x_t, t\right), \Sigma_\theta\left(x_t, t\right)\right)$$

(6)

Ho et al. (2020) suggests a simplification of Eq. 6 by employing a diagonal variance $\Sigma_\theta\left(x_t, t\right) = \sigma_t I$, where $\sigma_t$ are constants dependent on time. This narrows down the prediction task to $\mu_\theta\left(x_t, t\right)$. While a direct prediction of this term via a neural network seems the most intuitive, another method could involve predicting $x_0$ and then leveraging earlier equations to determine $\mu_\theta\left(x_t, t\right)$. Another possible method is inferring it by predicting the noise $\epsilon$. Indeed, (Ho et al., 2020) discovered that predicting the noise proved most effective. They propose the following parameterization:

$$\mu_\theta\left(x_t, t\right) = \frac{1}{\sqrt{\alpha_t}}\left(x_t - \frac{\beta_t}{\sqrt{1 - \bar{\alpha}_t}}\epsilon_\theta\left(x_t, t\right)\right)$$

(7)

where $\epsilon_\theta\left(x_t, t\right)$ is the prediction of the noise component $\epsilon$ used in the forward diffusion process between the timesteps $t - 1$ and $t$, and $\alpha_t := 1 - \beta_t$, $\bar{\alpha}_t := \prod_{i \leq t} \alpha_i$.

The objective Eq. 1 can be finally simplified to the sum of mean-squared errors between $\epsilon_\theta\left(x_t, t\right)$ and $\epsilon$ over all timesteps $t$:

$$L_{\text{simple}} = E_{t, x_0, \epsilon}\left[||\epsilon - \epsilon_\theta(x_t, t)||^2\right]$$

(8)

For a detailed derivation of these formulas and a deeper understanding of the methodologies, readers are encouraged to refer to the original paper (Ho et al., 2020; Nichol & Dhariwal, 2021).

**Multinomial diffusion models** (Hoogeboom et al., 2021) are designed to generate categorical data where $x_t \in \{0, 1\}^{Cl}$ is a one-hot encoded categorical variable with $Cl$ classes. In this case the aim of the forward Markov process is to convert the complex unknown data distribution into a known uniform distribution. The multinomial forward diffusion process $q\left(x_t|x_{t-1}\right)$ is a categorical distribution that corrupts the data by uniform noise over $Cl$ classes:

$$q\left(x_t|x_{t-1}\right) := Cat\left(x_t; (1 - \beta_t)x_{t-1} + \beta_t/Cl\right)$$

$$q\left(x_T\right) := Cat\left(x_T; 1/Cl\right)$$

$$q\left(x_t|x_0\right) := Cat\left(x_t; \bar{\alpha}_t x_0 + (1 - \bar{\alpha}_t)/Cl\right)$$

(9)

Intuitively, for each next timestep, a little amount of uniform noise $\beta_t$ over the $Cl$ classes is introduced, and with a large probability $(1 - \beta_t)$ the previous value $x_{t-1}$. From the equations above, the forward process posterior distribution $q(x_{t-1}|x_t, x_0)$ can be derived:

$$q\left(x_{t-1}|x_t, x_0\right) = Cat\left(x_{t-1}; \pi / \sum_{k=1}^{Cl} \pi_k\right)$$

(10)

where $\pi = [\alpha_t x_t + (1 - \alpha_t)/Cl] \odot [\bar{\alpha}_{t-1} x_0 + (1 - \bar{\alpha}_{t-1})/Cl]$.

The reverse distribution $p_\theta\left(x_{t-1}|x_t\right)$ is parameterized as $q\left(x_{t-1}|x_t, \hat{x}_0(x_t, t)\right)$, where $\hat{x}_0$ is predicted by a neural network. Specifically, in this approach, rather than directly estimating the noise component $\epsilon$, we predict $x_0$ which is then used to compute the reverse distribution. Then, the model is trained to maximize the variational lower bound Eq. 1.

## 4 TABGENDDPM

TabGenDDPM builds upon the principles of TabDDPM (Kotelnikov et al., 2023), primarily improving its capabilities in data imputation and synthetic data generation. The key distinctions lie in the denoising model and conditioning mechanism. While TabDDPM leverages a simple MLP architecture for

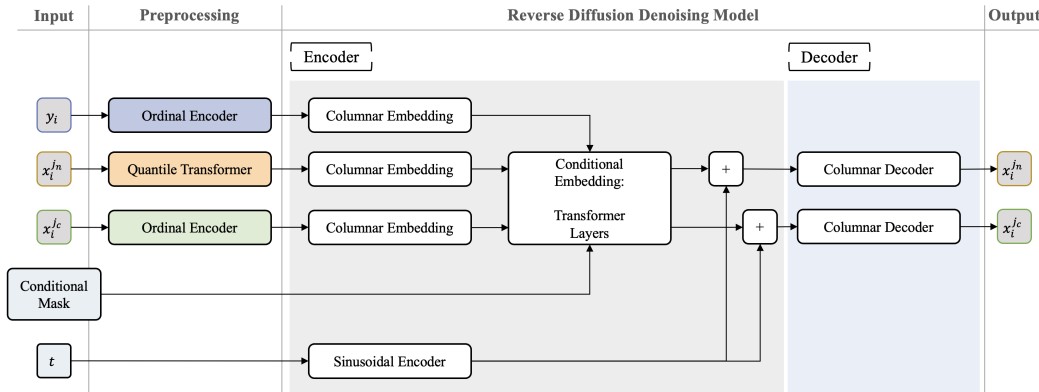

Figure 1: TabGenDDPM flow for the classification case

denoising, TabGenDDPM employs an encoder-decoder structure, introducing columnar embedding and *transformer* architecture. These modifications not only boost synthetic data quality but also offer improved conditioning for the reverse diffusion process. The resulting model is able to perform conditioned data generation and data imputation.

## 4.1 PROBLEM DEFINITION

We focus on tabular datasets for supervised tasks $D = \left\{ x_i^{j_c}, x_i^{j_n}, y_i \right\}_{i=1}^{N}$ where $x_i^{j_n}$ with $j_n \in [1, K_{num}]$ is the set of numerical features, $x_i^{j_c}$ with $j_c \in [1, K_{cat}]$ is the set of categorical features, $y_i$ is the label, $i \in [1, N]$ counts the dataset rows, $N$ is the total number of rows and $K = K_{num} + K_{cat}$ is total number of features.

We apply a consistent preprocessing procedure across our benchmark datasets, using the Gaussian quantile transformation from the scikit-learn library (Pedregosa et al., 2011) on numerical features and ordinal encoding for categorical ones. Missing values are replaced with zeros. In our approach we model numerical features with Gaussian diffusion and categorical features with multinomial diffusion. Each feature is subjected to a distinct forward diffusion procedure, which means that the noise components for each feature are sampled individually.

TabGenDDPM generalizes the approach of TabDDPM where the model learns $p\left(x_{t-1}|x_t, y\right)$, i.e. the probability distribution of $x_{t-1}$ given $x_t$ and the target $y$. We extend this by allowing conditioning on a target variable $y$ and a subset of input features, aligning with the strategies in Zheng & Charoenphakdee (2022) and Tashiro et al. (2021). Specifically, we partition variable $x$ into $x^M$ and $\bar{x}^M$. Here, $x^M$ refers to the masked variables set, those perturbed by the forward diffusion process, while $\bar{x}^M$ represents the untouched variable subset that conditions the reverse diffusion. This setup models $p\left(x_{t-1}^M|x_t^M, \bar{x}^M, y\right)$, with $\bar{x}^M$ remaining constant across timesteps $t$. This approach not only enhances model performance in data generation and but it also enables the possibility of performing data imputation with the same model.

The reverse diffusion process $p\left(x_{t-1}^M|x_t^M, \bar{x}^M, y\right)$ is parameterized by the neural network shown in Figs. 1 and 2. In the case of numerical features, the denoising model has to estimate the amount of noise added between steps $t-1$ and $t$ in the forward diffusion process, and in the case of categorical features, it must predict the (logit of) distribution of the categorical variable a $t = 0$. The model outputs has therefore dimensionality of $K_{num} + \sum_{i=1}^{K_{cat}} Cl_i$ where $Cl_i$ is the number of classes of $i$-th categorical feature.

## 4.2 MODEL OVERVIEW

The denoising model has an encoder-decoder structure as show in Fig. 1. The encoder obtains a representation of each features in two step: first a columnar embedding individually projects all the heterogeneous features (continuous or categorical) in the same homogeneous and dense latent space.

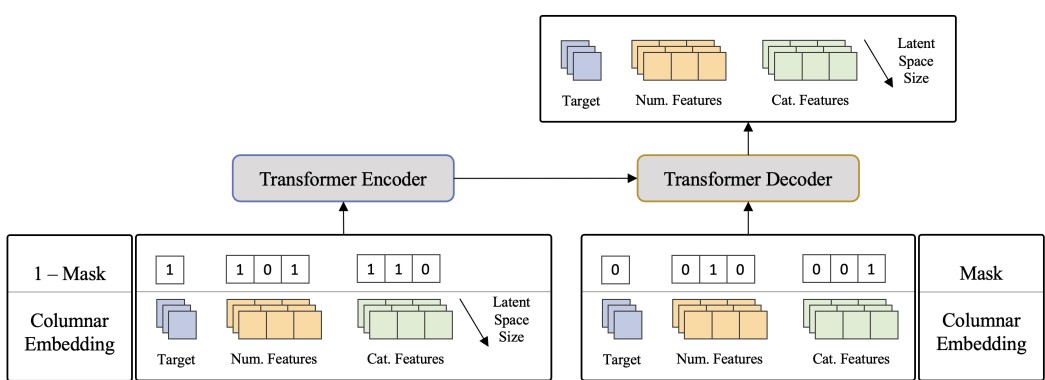

Figure 2: Conditional Transformer Encoder

Then a conditional transformer embedding enhances the features latent representation by accounting for their inter-feature interaction.

For categorical and numerical features, we utilize an Embedding layer and a linear-ReLU activation layer, respectively. The type of task (regression or classification) dictates the choice of embedding for the target, $y$.

The conditional transformer (detailed in Fig. 2) incorporates columnar embeddings through specialized masking. With this, the encoder's attention mask focuses on latent embeddings of variables $\bar{x}^M$ and $y$, while the decoder attends to $x^M$ embeddings. In this setup, the encoder output provides context from variables not involved in the diffusion, allowing the decoder to derive representations for those that are. Features involved in the forward diffusion process are denoted with $\texttt{Mask} = 1$, and those that aren't with $\texttt{Mask} = 0$. Consequently, the encoder takes in $1 - \texttt{Mask}$ while the decoder works with $\texttt{Mask}$.

The final latent feature representation is obtained by summing the conditional transformer embedding output with the timestep embedding, which is derived by projecting the sinusoidal time embedding (Nichol & Dhariwal, 2021; Dhariwal & Nichol, 2021), into the transformer embedding dimension using a linear layer followed by the Mish activation function (Misra, 2020).

Finally, this representation is decoded to produce the output. Each feature has its own decoder consisting of two successive blocks integrating a linear transformation, normalization, and ReLU activation. Depending on the feature type (numerical or categorical), an additional linear layer is appended with either a singular output for numerical features or multiple outputs corresponding to the number of classes for categorical ones.

The model is trained by minimizing a sum of mean-squared error $L_t^{simple}$ (Eq. 8) for the Gaussian diffusion term and the KL divergences $L_t^i$ for each multinomial diffusion term (Eq. 1).

$$L_t^{TabGenDDPM} = \frac{L_t^{simple}(\texttt{Mask})}{K_{num}} + \frac{\sum_{i \leq K_{cat}} L_t^i(\texttt{Mask})/Cl_i}{K_{cat}} \tag{11}$$

where $L_t^{simple}(\texttt{Mask})$ and $L_t^i(\texttt{Mask})$ means that the loss functions are computed taking into account only the prediction error on variables affected by the forward diffusion process (i.e. $x^M$), $Cl_i$ is the number of classes of the $i$-th categorical variable.

### 4.3 DYNAMIC CONDITIONING

A key feature of the current proposal is that the split between $x^M$ and $\bar{x}^M$ does not have to be fixed for all the row $i$ in the dataset. Transformer encoder and decoder can manage mask with an arbitrary number of zeros/ones, so we can dynamically alter the split between $x^M$ (decoder input) and $\bar{x}^M$ (encoder input) by just producing a new mask. In the extreme scenario, we can generate a new mask $\texttt{Mask}_i$ for each row $i$. During training, the number of ones in $\texttt{Mask}_i$ (i. e. the number of features to be included in the forward diffusion process) is uniformly sampled from the interval $[1, M_{num} + M_{cat}]$. A model that has been trained in this manner may then be used for both,

generation of synthetic data ($\texttt{Mask}_i = 1$ for all the $M_{num} + M_{cat}$ features and for any dataset index $i$) and imputation of missing values (for each $i$, $\texttt{Mask}_i = 1$ for the feature to impute).

This setup allows for more flexible conditioning scenarios. Specifically:

- When $\bar{x}^M = \emptyset$, $p\left(x_{t-1}^M \middle| x_t^M, \bar{x}^M, y\right) = p\left(x_{t-1} \middle| x_t, y\right)$ our model aligns with TabDDPM, generating synthetic data influenced by the target distribution.
- When $\bar{x}^M \neq \emptyset$, the model can generate synthetic data based on the target distribution and either a fixed or dynamic subset of features. Conditioning on a fixed subset is particularly beneficial in situations where certain variables are easily accessible, while others are harder to obtain due to challenges like pricing. In these cases, the scarce data can be synthetically produced using the known variables. Conversely, when conditioning on a dynamic subset of features, the model effectively addresses the challenge of imputing gaps within a dataset.

## 5  EXPERIMENTS

### 5.1  DATA

The benchmark used to evaluate the model is summarized in Table 1. Detailed descriptions of the datasets can be found in the Appendix A.

Table 1: Tabular benchmark properties. In the case of categorical features, numbers within parenthesis indicate the number of categories for each categorical features.

| Dataset | Rows | Num. Feats | Cat. Feats | Task |
|---|---|---|---|---|
| HELOC | 9871 | 21 | 2 | Binary |
| Churn | 10000 | 6 | 4 | Binary |
| Cal. Hous. | 20640 | 8 | 0 | Regression |
| House Sales | 21613 | 14 | 2 | Regression |
| Adult Inc. | 32561 | 6 | 8 | Binary |
| Cardio | 70000 | 7 | 4 | Binary |
| Insurance | 79900 | 8 | 2 | Binary |
| Forest Cov. | 581 K | 10 | 2 | Multi-Class (7) |

### 5.2  BASELINES

We have selected to evaluate against the foremost representatives from each generative modeling paradigm, being VAE, GAN or Diffusion Models:

*TabDDPM* (Kotelnikov et al., 2023): state-of-the-art diffusion model for tabular data generation.
*TVAE* (Xu et al., 2019): a variational autoencoder adapted for mixed-type tabular data.
*CTGAN* (Xu et al., 2019): a conditional GAN for synthetic tabular data generator.

### 5.3  METRICS

We evaluate the generative models on three different dimensions: 1) ML efficiency, 2) statistical similarity and 3) privacy risk.

The *Machine Learning efficiency* measures the performance degradation of classification or regression models trained on synthetic data and then tested on real data. The basic idea is to use a ML discriminative model to evaluate the quality of synthetic data provided by a generative model. As established by Kotelnikov et al. (2023), a strong ML model allows to obtain more stable and consistent conclusions on the performances of the generative model. For this reason, as a first step, we use the Optuna library (Akiba et al., 2019) aiming at training the best possible discriminative model. Optuna is run for 100 iterations to fine-tune the XGBoost (Chen & Guestrin, 2016) hyperparameters on each dataset's real data within the benchmark. Every hyperparameter configuration for XGBoost is crossvalidated with five folds. The complete hyperparameter search space is depicted in appendix B Table 5.

Once the discriminative model has been optimized for each dataset, the generative model is cross-validated with five folds using the following procedure. For each fold, the real data is splitted in

three subsets. The main aim of the first subset is training the generative model. The resultant model generates a synthetic dataset conditioned by the second subset. The synthetic dataset is then used to train the discriminative model. The so-obtained XGBoost is finally tested on the third subset that has never been seen during the training of any model. The procedure is repeated for each fold and the obtained metric mean is used as a final measure of the generative model ML utility.

*Statistical similarity.* The comparison between synthetic and real data considers both individual and joint feature distributions. Adopting the approach described by Zhao et al. (2021), we employ Wasserstein Wang et al. (2021a) and Jensen-Shannon distances Lin (1991) to analyze numerical and categorical distributions. Additionally, the square difference between pair-wise correlation matrix is used to evaluate the preservation of feature interactions in synthetic datasets. Specifically, the Pearson correlation coefficient measures correlations among numerical features, the Theil uncertainty coefficient measures correlations among categorical features, and the correlation ratio evaluates interactions between numerical and categorical features.

The *Privacy Risk* is evaluated using the Distance to Closest Record (DCR), i.e. the Euclidean distance between any synthetic record and its closest corresponding real neighbour. Ideally, the higher the DCR the lesser the risk of privacy breach. It is important to underline that out-of-distribution data, i.e. random noise, will also provide high DCR. Therefore, DCR needs to be evaluated along with ML efficiency together.

## 6 RESULTS

**Machine Learning efficiency - Synthetic data generation**

The task of the generative model in this context is to produce synthetic data, conditioned exclusively by the supervised target $y$. For clarity and to facilitate the comparison of results, two models from our suite are discussed:

1. *TabGenDDPM I*: This model consistently includes all dataset features in the diffusion process. It is specifically designed for this use-case.

2. *TabGenDDPM II*: During training, this model dynamically selects which features are incorporated in the diffusion process, making it versatile for both imputing missing data and generating complete synthetic datasets.

The results, summarized in Tab. 2, demonstrate that when TabGenDDPM II is used for synthetic data generation, it outperforms existing literature methods like TabDDPM, TVAE, and CTGAN. Only the specialized TabGenDDPM I achieves superior performance in this scenario of data generation. The key outcomes of our experiments are as follows: 1) TVAE produces better results than CTGAN, 2) TabDDPM outperforms TVAE and CTGAN in mean, and 3) Our two proposed models systemati-

Table 2: *Machine Learning efficiency* or *utility*. Classification tasks use F1-score, and regression tasks use MSE, indicated by up/down arrows for maximization/minimization of the metric. Cross-validation mean and standard deviation are shown for each dataset-model pair. Best and second-best results are highlighted in bold and underline, respectively. **Baseline** column shows XGBoost performance trained on real data, while other columns reflect XGBoost trained on synthetic data from specified models. All models are tested on real data.

| Dataset | Baseline | TVAE | CTGAN | TabDDPM | TabGenDDPM I | TabGenDDPM II |
|---|---|---|---|---|---|---|
| HELOC ↑ | $83.72 \pm 0.70$ | $79.40 \pm 1.10$ | $77.54 \pm 0.77$ | $76.66 \pm 0.49$ | $\mathbf{83.12 \pm 0.61}$ | $\underline{82.51 \pm 0.58}$ |
| Churn ↑ | $85.29 \pm 0.37$ | $81.68 \pm 0.69$ | $79.33 \pm 1.10$ | $83.57 \pm 0.56$ | $\mathbf{84.03 \pm 0.39}$ | $\underline{83.69 \pm 0.38}$ |
| Cal. Hous. ↓ | $0.147 \pm 0.008$ | $0.316 \pm 0.001$ | $0.488 \pm 0.030$ | $0,272 \pm 0.009$ | $\mathbf{0.214 \pm 0.004}$ | $\underline{0.226 \pm 0.005}$ |
| House Sales ↓ | $0.095 \pm 0.008$ | $0.209 \pm 0.030$ | $0.335 \pm 0.022$ | $0.145 \pm 0.014$ | $\mathbf{0.121 \pm 0.006}$ | $\underline{0.141 \pm 0.005}$ |
| Adult Inc. ↑ | $86.99 \pm 0.18$ | $84.35 \pm 0.80$ | $83,63 \pm 0.40$ | $84.81 \pm 0.20$ | $\mathbf{85.30 \pm 0.20}$ | $\underline{85.10 \pm 0.15}$ |
| Cardio ↑ | $73.74 \pm 0.14$ | $72.49 \pm 0.35$ | $71.8 \pm 0.45$ | $72.83 \pm 0.25$ | $\underline{72.96 \pm 0.24}$ | $\mathbf{73.07 \pm 0.19}$ |
| Insurance ↑ | $91.99 \pm 0.25$ | $92.62 \pm 0.26$ | $92.55 \pm 0.19$ | $92.2 \pm 0.30$ | $\mathbf{92.76 \pm 0.12}$ | $\underline{92.65 \pm 0.14}$ |
| Forest Cov. ↑ | $96.54 \pm 0.08$ | $70.35 \pm 0.67$ | $65.63 \pm 0.35$ | $82.08 \pm 0.19$ | $\mathbf{84.15 \pm 0.39}$ | $\underline{83.25 \pm 0.33}$ |

cally enhances the performance of TabDDPM, attaining the best results across all datasets included in our benchmark.

It is worth noting that the size of the generated samples matches the size of the real data for comparison purposes. However, our preliminary results show that the results smaller datasets could be improved if the size of synthetic data surpasses the original data size. To obtain the results presented in Table 2, each generative model is optimized with Optuna over 100 trials, with the cross-validated ML efficiency defined in Section 5 as the objective. The specific hyperparameters search space for each model is shown in Table 6 in appendix B.

**Machine Learning efficiency - Data imputation**

These experiments are aimed to evaluate the model ability to impute missing values. In this context, the generative model utilizes the available data to condition the generation of data for the missing entries. Assessing the quality of imputed data using ML utility, which is based on XGBoost performance, requires careful consideration. This is because not all features equally influence the XGBoost's output. The impact of imputing a highly significant feature versus one with minimal impact on XGBoost's performance can vary greatly. To address this, we propose a randomized approach for feature selection to be imputed: 1) fix the number of required missing values, 2) using uniform distribution, pick the features with missing value (i.e change their values to None) for each row, 3) use the generative model to fill in the missing values, 4) train the XGBoost on the imputed data 5) evaluate the XGBoost over the real hold-out data 6) Repeat steps 1-5 increasing the amount of missing from one to $M_{num} + M_{cat}$.

Our experiments, depicted in Fig. 3, compares the performance of our model compared to a modified version of TabDDPM such that it may be conditioned by any arbitrary subset of features. We also incorporate two additional baselines: missForest (Stekhoven & Stekhoven, 2013) and GAIN (Yoon et al., 2018). Unlike TabDDPM and TabGenDDPM, which are trained once with a number of missing values uniformly distributed from one to $M_{num} + M_{cat}$ and then applied to various missing data scenarios, MissForest and GAIN undergo training and imputation separately for different levels of missing features. Consequently, they tend to underperform when faced with more than $50\%$ missing data. In contrast, both TabDDPM and TabGenDDPM outperform these baselines across all levels of missing data, with TabGenDDPM having the best performance, especially as the number of missing data grows.

**Statistical Similarity**

The summary of results obtained over our benchmark are shown in Tab. 3. It show the average ranks computed over all datasets in the benchmark: lower is better. The Wasserstein distance is used to calculate rank for numerical features whereas the Jensen–Shannon distance is used for the categorical ones. Finally the L2 distance between correlation matrices is employed to assess how effectively feature interactions are retained in synthetic datasets. Distances are calculated between synthetic data and real data.

Table 3: *Statistical Similarity* between synthetic data and real data.

|  | TabDDPM | TabGenDDPM |
|---|---|---|
| Wasserstein Distance | 1.84 | 1.16 |
| Jensen-Shannon Distance | 1.60 | 1.40 |
| L2 Dist. Correlation Matrix | 2.00 | 1.00 |

**Privacy risk**

In terms of privacy risk, we believe the results displayed in Table 4 are promising. The ML efficiency is consistently superior, and the generated data more closely follows the original distribution, as indicated by the L2 distances of the correlation matrix, as well as the Wasserstein and Jensen-Shannon distances. This increased fidelity contributes to the slightly elevated risk. An example of this is the HELOC dataset, where the risk is higher than the baseline which is produced by a significant improvement in ML efficiency. Moreover, we have verified that none of the synthetically generated samples have a distance of zero from the original samples. If additional privacy measures are necessary, it is possible to incorporate *differential privacy* into the generative model (Jälkö et al.,

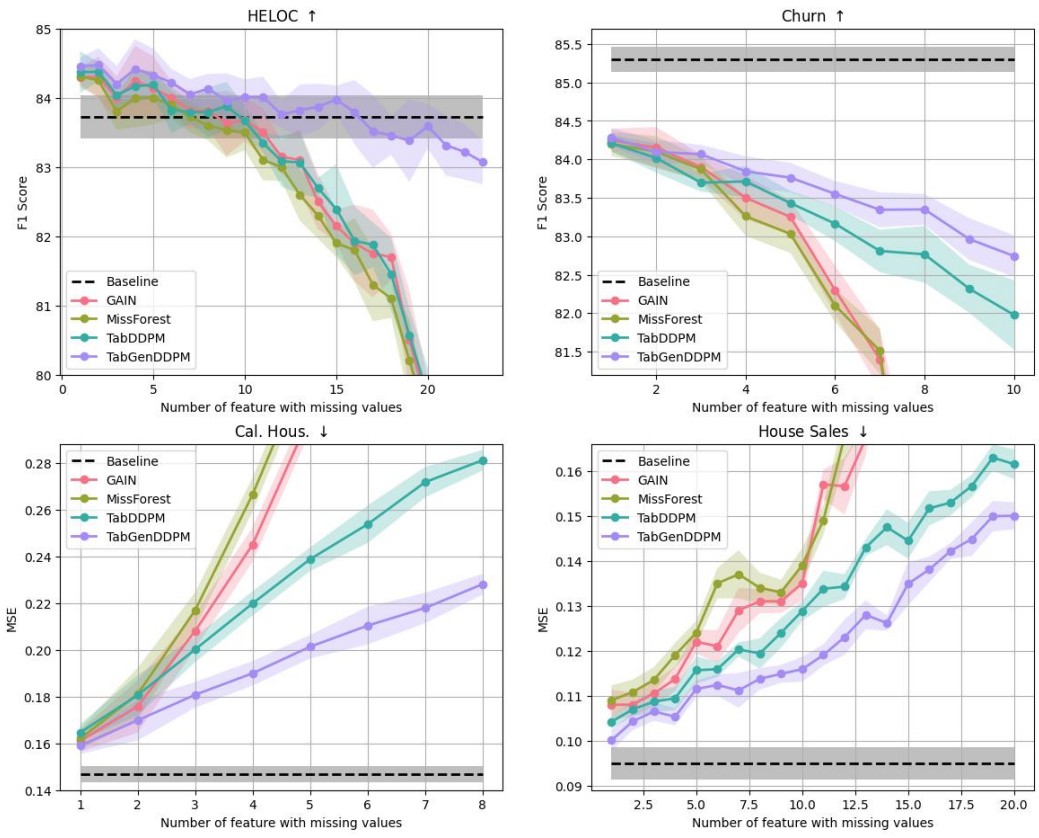

Figure 3: Results for data imputation use case in terms of ML utility under different levels of missing features. Baseline is an oracle that imputes real data.

2021). This would allow us to control the trade-off between the utility of the generated data and the associated privacy risk.

Table 4: Comparison of correlation matrix L2 distances, privacy risk and ML efficiency ( ↑ for F1-score, ↓ for MSE) for TabDDPM and TabGenDDPM for each dataset. Privacy risk is evaluated using the Distance to Closest Record and higher the this value, the lesser the risk of privacy breach.

| | **TabDDPM** | | | **TabGenDDPM** | | |
|---|---|---|---|---|---|---|
| | Corr. | Risk | ML eff. | Corr. | Risk | ML eff. |
| HELOC ↑ | 0.03 | 2.75 | 76.66 | 0.005 | 0.25 | 82.7 |
| Churn ↑ | 0.0013 | 0.09 | 83.57 | 0.0008 | 0.09 | 83.98 |
| Cal. Hous. ↓ | 0.211 | 1.28 | 0.27 | 0.0018 | 0.075 | 0.21 |
| House Sales ↓ | 0.018 | 0.34 | 0.145 | 0.003 | 0.1 | 0.131 |
| Adult Inc. ↑ | 0.01 | 0.15 | 84.8 | 0.0035 | 0.11 | 85.3 |
| Cardio ↑ | 0.0035 | 0.41 | 72.83 | 0.0022 | 0.41 | 72.96 |

## 7 CONCLUSION

In our exploration of synthetic tabular data generation and data imputation, we introduced a novel adaptation to TabDDPM diffusion model, incorporating a transformer and unique masking mechanism for conditioning the reverse diffusion process. This innovation allows our model to handle both tasks within a unified framework. Our evaluations show our model's better performance over the baselines for synthetic data generation based on VAE, GAN and diffusion model, in terms of ML utility, statistical accuracy while keeping similar privacy risk. This study, thus, not only bridges the conceptual gap between data imputation and synthetic data generation but also sets a new benchmark for generative models in tabular data contexts.

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

## Appendices

## A   Dataset Descriptions

*HELOC* FICO (2019): Home Equity Line of Credit (HELOC) provided by FICO (a data analytics company), contains anonymized credit applications of HELOC credit lines. The dataset contains 21 numerical and two categorical features characterizing the applicant to the HELOC credit line. The task is a binary classification and the goal is to predict whether the applicant will make timely payments over a two-year period.

*Churn Modelling* Iyyer (2019): This dataset consists of six numerical and four categorical features about bank customers. The binary classification task involves determining whether or not the customer closes his account.

*California Housing* Pace & Barry (1997): The information refers to the houses located in a certain California district, as well as some basic statistics about them based on 1990 census data. This is a regression task, which requires to forecast the price of a property.

*House Sales King Country* Kaggle (2016): Similar to the California Housing case, this is a regression task in which the prices of properties sold in the King County region between May 2014 and May 2015 must be estimated. The dataset originally contained 14 numerical and four categorical features, as well as one date. The date is turned into two categorical variables (month and year) after our pre-processing.

*Adult Incoming* Becker & Kohavi (1996): Personal details such as age, gender or education level are used to predict whether an individual would earn more or less than 50K\$ per year.

*Cardiovascular Disease* Ulianova (2020): The existence or absence of cardiovascular disease must be predicted based on factual information, medical examination results, and information provided by the patient. The dataset is made up of seven numerical and four categorical features.

*Insurance* Rathi & Mishra (2019) Customer variables and past payment data are used to solve a binary task: determining whether the customer will pay on time. There are eight numerical and two categorical features in the dataset.

*Forest Cover Type* Blackard (1998): Cartographic variables are used to predict the forest cover type: it is a multi-class (seven) classification task. The first eight features are continuous whereas the last two are categorical with four and 40 levels, respectively.

## B   Hyperparameter tuning

Table 5: Search Space for XGBoost Hyperparameters with Optuna.

| Parameter | Range |
|---|---|
| max depth | $[1, 9]$ |
| learning rate | $[0.01, 1.0]$ |
| estimators | $[50, 500]$ |
| min child weight | $[1, 10]$ |
| gamma | $[10^{-8}, 1]$ |
| subsample | $[0.01, 1]$ |
| colsample bytree | $[0.01, 1]$ |
| reg alpha | $[10^{-8}, 1]$ |
| reg lambda | $[10^{-8}, 1]$ |

Table 6: Hyperparameters for Different Models.

| Model | Hyperparameter | Possible Values |
|---|---|---|
| TVAE | compress dims | $[32, 64, 128, 256, 512]$ |
| | decompress dims | $[32, 64, 128, 256, 512]$ |
| | embedding dim | $[32, 64, 128, 256, 512]$ |
| | batch size | $[64, 128, 256, 512, 1024]$ |
| | epochs | $500$ |
| CTGAN | generator dim | $[32, 64, 128, 256, 512]$ |
| | discriminator dim | $[32, 64, 128, 256, 512]$ |
| | embedding dim | $[32, 64, 128, 256, 512]$ |
| | batch size | $[64, 128, 256, 512, 1024]$ |
| | epochs | $500$ |
| TabDDPM | timesteps | $[100, 200, 300, 400, 600, 800, 1000]$ |
| | latent space size | $[64, 128, 256, 512, 1024]$ |
| | mlp depth | $[2, 4, 6, 8]$ |
| | batch size | $[64, 128, 256, 512, 1024]$ |
| | epochs | $500$ |
| TabGenDDPM | timesteps | $[100, 200, 300, 400, 600, 800, 1000]$ |
| | latent space size | $[64, 128, 256, 512, 1024]$ |
| | transformer layer num | $[2, 3, 4]$ |
| | transformer heads | $[2, 4, 8]$ |
| | transformer feedforward size | $[256, 512]$ |
| | batch size | $[64, 128, 256, 512, 1024]$ |
| | epochs | $500$ |

