# OpenReview forum: "Diffusion Models for Tabular Data Imputation and Synthetic Data Generation"
_ICLR.cc/2024/Conference — Submitted to ICLR 2024_

### Official Review · Reviewer_Cv2X · 2023-10-31

**Soundness:** 3 good
**Presentation:** 3 good
**Contribution:** 3 good
**Rating:** 6
**Confidence:** 3

**Summary:**

The authors introduce a we a novel adaptation to TabDDPM diffusion model, incorporating a transformer (compared to MLP for TabDDPM) and unique masking mechanism to condition the reverse diffusion process. This encoder-decoder structure, allows for introducing columnar embedding and enables data imputation as well as data conditioning. Empirical results seem to support the new model, with better ML utlity at the cost of higher risk of privacy breach.

**Strengths:**

- Well written paper, with clear figures, no grammatical issues, and good flow
- Empirical results can be directly compared to previous baselines
- Novelty is clear and well explained
- Datasets and baselines are appropriate for the evaluation task

**Weaknesses:**

- It would be nice to see a few more plots of the feature distribution rather than a simple distribution difference score
- Analysis of method on the same 15 datasets as the reference TabDDPM paper would be useful
- Further ablations / discussions showing the imputation would also add to the paper. For example, why is the performance worse with  TabGenDDPM I vs II?

**Questions:**

- For ML efficiency, the original TabDDPM paper demonstrate multiple examples where the generated data is able to achieve better performance over the baseline. However, this behavior is not seen here?
- Is it possible to conditionally generate diverse synthetic data by conditioning on an outcome feature? E.g. death event or housing price?
- Does this model faithfully generate data that captures low-domain clusters / phenotypes in the original data space?
- Why does the best performance switch for TabGenDDPM I and II in the cardio dataset?

---

> ### Author Response · Authors · 2023-11-22
>
> Dear reviewer Cv2X,
>
> Thank you so much for taking the time to review our submission. We will try to address all questions:
>
>
> **Q**: For ML efficiency, the original TabDDPM paper demonstrate multiple examples where the generated data is able to achieve better performance over the baseline. However, this behavior is not seen here?
>
> **A**: In our study, we defined the baseline as an oracle that either generates or imputates the real data. We have clarified this in the updated caption of Table 2 in the results section. Our baseline is the performance of XGBoost trained and tested on real data, representing the optimal scenario. In the case of TabDDPM they also compare training with the real data and their model outperforms that baselines for 4 datasets, while in our case we outperform it for the Insurance dataset only.
>
> One key difference between TabDDPM and ours is in the method used for measuring the ML efficiency. While we use the exact same number of samples as the original training data, TabDDPM employs a “proportion of samples” hyper-parameter that can lead to generating up to eight times the volume of real samples for certain datasets. This difference in the volume of generated samples could potentially account for the observed variations in performance relative to the baseline.
>
> **Q**: Is it possible to conditionally generate diverse synthetic data by conditioning on an outcome feature? E.g. death event or housing price?
>
> **A**: Yes, it is possible to generate synthetic data conditioned on any feature.
>
>
> **Q**: Does this model faithfully generate data that captures low-domain clusters / phenotypes in the original data space?
>
> **A**: Our model is designed to capture and replicate the complex distributions and relationships present in the original data. However, the exact fidelity in representing low-domain clusters or specific phenotypes would require further detailed analysis, which could be a direction for future research.
>
> **Q**: Why does the best performance switch for TabGenDDPM I and II in the cardio dataset?
>
> **A**:  The observed performance variation between the models is interesting and leads us to consider several hypotheses. First, we hypothesize that in the case of the Cardio dataset, the discriminative power of the XGBoost model is somewhat limited, as indicated by its relatively lower performance even when trained with real data (baseline). This limitation introduces a level of noise to the results, potentially influencing the observed switch in performance ranking between TabGenDDPM I and II.
>
> A second hypothesis considers the characteristics of the cardio dataset, which contains features that are highly correlated. The dynamic conditioning mechanism in our model, which varies the masked features during training, could potentially disrupt these correlations and may contribute to training a more robust model.
>  Although these hypotheses may provide possible explanations, definitive understanding of the performance switch would need a deeper analysis.

---

### Official Review · Reviewer_AvR7 · 2023-10-31

**Soundness:** 3 good
**Presentation:** 3 good
**Contribution:** 1 poor
**Rating:** 6
**Confidence:** 4

**Summary:**

This paper propose an improvement of TabDDPM model through the addition of tree improvements:
- 1. The categorical columns are encoded via an Embedding layer instead of a one-hot and the numeric columns are encoded through a linear layer. This allows an uniform encoding of the columns into the same dimension independently of their type.
- 2. The MLP denoiser of TabDDPM is replaced by a transformer architecture.
- 3. A BERT-like attention masking system is then used to train the model for dynamic conditioning and missing data imputation

After quick introduction and related work sections, a background is given about ddpm and multinomial diffusion algorithms.
Then the specificities of the model are presented.
The experiment section compare two variants of the model against TVAE, CTGAN and vanilla TabDDPM (all with optimized hyper-parameters) on 7 datasets through ML-efficacy and DCR privacy risk. Another statistical similarity metric is proposed as well.
The two variants considered are trained with full data (TabGenDDPM I) and with masked data (TabGenDDPM II).
For ML-efficacy TabGenDDPM I is shown to outperform the other models on 6 out of the 7 selected datasets. For data imputation TabGenDDPM is also shown to outperform a customized version of TabDDPM.
On the other hand, the privacy risk is reported to be slightly higher than with TabDDPM.

**Strengths:**

- The proposed architecture is a natural improvement from TabDDPM and according to the experiments, it seems to really improve the model in term of ML-efficacy
- The paper is clear and well written with several illustrations
- The privacy risk is considered

**Weaknesses:**

- The proposed architecture is mostly a derivative work from TabDDPM
- The proposed diffusion algorithms are a bit outdated now, especially on the discrete side since works like:
Austin et al. "Structured Denoising Diffusion Models in Discrete State-Spaces" NeurIPS 2021, or Campbell et al. "A Continuous Time Framework for Discrete Denoising Models" NeurIPS 2022.
It is worth noting that "mask" systems are also studied in (Austin et al. 2021).
- No ablation study to validate the separately different changes from TabDDPM (eg. category embedding vs one-hot)
- No simple "non-deep" baseline model (like SMOTE) in the experiment.
- The code seems not to be open source

**Questions:**

- The hyper-parameter space of TabDDPM seems modified in your experiment (e.g. no batch size 4096 and no learning rate) Why ?
- With the masking system it is possible to condition on any feature. Why keep a specific treatment for the target value ?
- The statistical similarity metric is not usual and do not permits an easy comparison with other papers, why not use "sdmetrics" library to provide other metrics (notably C2ST detection metrics) ?

---

> ### Author Response · Authors · 2023-11-22
>
> Dear Reviewer AvR7,
>
> We sincerely thank you for your thoughtful comments and suggestions. We appreciate the time and effort you have invested in reviewing our paper. Below, we try address each of the concerns and comments you have provided:
>
>
>
> **Q**: The hyper-parameter space of TabDDPM seems modified in your experiment (e.g. no batch size 4096 and no learning rate) Why ?
>
> **A**: You're correct in observing the differences in hyper-parameter space. Our decision was driven by the results of preliminary hyper-parameter optimization using Optuna, which indicated superior performance with smaller batch sizes, hence the absence of batch sizes above 1024 in our experiments. Additionally, given the constraints of our computational resources, we had to limit the scope of our search space.
>
> **Q**: With the masking system it is possible to condition on any feature. Why keep a specific treatment for the target value ?
>
> **A**: This is an insightful observation. Our decision to treat the target variable distinctly and keep it always unmasked was primarily motivated by our focus on generating synthetic samples conditioned on the target.  This approach aligns with the class-conditional model used by TabDDPM,
>
> **Q**: The statistical similarity metric is not usual and do not permits an easy comparison with other papers, why not use "sdmetrics" library to provide other metrics (notably C2ST detection metrics) ?
>
> **A**: Thank you for this suggestion. Our choice of statistical similarity metrics was influenced by precedents in related works, such as TabDDPM and CTAB-GAN [1], which utilize correlation similarity, Wasserstein, and Jensen-Shannon distances. However, we appreciate the relevance of the "sdmetrics" library and we certainly plan to incorporate these in future revisions of our work or subsequent studies to provide a more comprehensive and comparable analysis.
>
> [1] Zhao, Zilong, et al. "Ctab-gan: Effective table data synthesizing." Asian Conference on Machine Learning. PMLR, 2021.
>
> **C** Code is not present.
>
> **A** Thank you for pointing this out. To maintain the integrity of the double-blind review process, we've included the code as supplementary material in a zip file. We plan to publish it in github after the review process is over.

---

### Official Review · Reviewer_uxeP · 2023-11-01

**Soundness:** 2 fair
**Presentation:** 2 fair
**Contribution:** 2 fair
**Rating:** 5
**Confidence:** 5

**Summary:**

The paper proposes a transformer conditioning architecture design on TabDDPM for data imputation and data generation tasks. They conduct experiments on eight datasets under machine learning utility, statistical similarity, and privacy risk.

**Strengths:**

The experimental comparisons are good. The author conducts TabGenDDPM on eight datasets under three evaluation criteria.

**Weaknesses:**

1. The overall contribution of this paper is limited.

All of the content except the transformer conditioning architecture is already known. The architecture design is heuristic, which has no theoretical guarantees of the performance. Moreover, they build upon Variance Preserving (VP) SDE (e.g., DDPM or TabDDPM in tabular data). The author does not mention wether their method work for Variance Exploding (VE) SDE (e.g, Score-based generative model, StaSy [1] in tabular data).
[1]: Kim, J., Lee, C.E., & Park, N. STaSy: Score-based Tabular data Synthesis. ICLR 2023.

2. Overclaiming the contribution of transformer conditioning architecture.

* Diffusion model can work on imputation together with generation (conditional generation) without the proposed transformer conditioning architecture. There are well studied in the literature [1,2].

[1]: Tashiro, Y., Song, J., Song, Y., & Ermon, S. CSDI: Conditional Score-based Diffusion Models for Probabilistic Time Series Imputation. NIPS 2021.

[2]: Ouyang, Y., Xie, L., Li, C., & Cheng, G. (2023). MissDiff: Training Diffusion Models on Tabular Data with Missing Values. ArXiv, abs/2307.00467.

3. The effectiveness of the proposed method is not well supported.
* : The standard evaluation of imputation performance is the mean squared error of imputed value against oracle value instead of the efficiency criterion used in paragraph "Machine Learning efficiency - Data imputation". Otherwise, it faces the problem of "when the generative model needs to fill in the most significant feature or a feature that has a minimal impact on XGBoost output" mentioned in the paper. If the authors adopt the traditional evaluation on this task, many design in this paragraph will not be needed.

* : To evaluate the performance of TabGenDDPM on imputation task, it should be compared with other imputation methods, e.g., [3,4], rather than only compared with TabDDPM.

[3]: Yoon, J., Jordon, J., & Schaar, M.V. GAIN: Missing Data Imputation using Generative Adversarial Nets. ICML 2018.

[4]: Mattei, P., & Frellsen, J. MIWAE: Deep Generative Modelling and Imputation of Incomplete Data Sets. ICML 2019.


* : The author should compare with other diffusion based model on tabular data, e.g., StaSy [1]. Also, some discussion and experimental results of whether transformer conditioning can  be developed on Variance Exploding (VE) SDE.

* : The of illumination the experimental setup should be clarify. Currently, it brings some confusion.
- The baseline in Figure 3 stands for which method? In my point of view, it is not the methods mentioned in section 5.2.
- The Table 4 is confusion. In my point of view, three different evaluation criteria have different properties, i.e., the smaller the correlation is, the better the performance is, which is different with privacy risk. Why the authors use Up arrow/Down arrow beside the name of the dataset. It is also not clear why the authors only report the experimental results on six datasets rather than eight datasets in Table 2.
- It would be helpful to have the performance on each dataset for Table 3 in appendix.

4. Minor

The paper has many typos, e.g.,
- adding period for the caption of Table 1, 3, 4 and Figure 3;
- what is the meaning of "4+2" and "2(4+40)" in Table 1;
- "in this situation, the generative model can employ the no-missing values to condition the missing data generation." is hard to understand.

**Questions:**

Please see Weaknesses Part.

---

> ### Author Response · Authors · 2023-11-22
>
> Dear Reviewer uxeP,
>
>
> Thank you for your insightful comments and for bringing the works involving Variance Exploding SDE like StaSy. We acknowledge that including a comparison with StaSy would have enriched our study. At the time of our research and manuscript preparation, we were not aware of this work and the application of VE SDE to tabular data. We plan to adapt the model to VE SDE and compare with other Score-based tabular generative models in future revisions of this work or subsequent studies.
>
> Regarding other works that perform data imputation together with generation, the first paper CSDI only works with continuous variables in temporal series, and it is not straightforward of how to adapt it for tabular data. The second proposed paper, MissDiff, rises a broader discussion whether considering preprint papers under review for same conference. MissDiff is also under review for ICLR 2024 https://openreview.net/forum?id=vULHgaoASR, which we will consider for future research.
>
>
> Regarding the other comments, we are grateful for your valuable feedback and helping us improve the paper.
>
> * We have used ML efficiency for evaluating the data imputation task, because the imputed data doesn't need to be exactly the same as the original data, but just follow the same distribution conditioned on the available data. In our opinion, ML efficiency measures the imputation performance better than MSE of imputed data with real data. In addition to that, there are also categorical variables in the datasets, where the MSE is not as descriptive as in continuous variables.
> * We have added two baselines for the data imputation task and updated the graphs and the text under  “Data imputation” in the section “Results”. The baselines are GAIN and missForest. We didn't include MIWAE, because after carefully analyzing the code available we think it wouldn't be easy to adapt to categorical variables.
> * We have also updated the caption in Table 2 and Figure 3. Regarding the baseline here, it is the same XGBoost trained on original real data and tested on real data, and the scores of the baseline stand as what would be the best score in optimal conditions. Nevertheless, for the “Insurance” dataset all tested methods surpass this oracle baseline.
> * In Figure 3. The baseline is an oracle data imputation that imputes real data and that its performance is the same as in Table 2. We have updated the table to make it clear.
> * In table 4, we have changed the caption to make clear that the “up arrow” stands for a maximization optimization measuring the ML efficiency in F1 score and a “down arrow” for a minimization optimization of the MSE.
>
> We are grateful for your comments, which have significantly contributed to the improvement of our paper.

---

> > ### Comment · Reviewer_uxeP · 2023-11-23
> >
> > Thank you very much for the detailed responses. My overall rating remains.

---

### Official Review · Reviewer_TceJ · 2023-11-05

**Soundness:** 3 good
**Presentation:** 3 good
**Contribution:** 3 good
**Rating:** 6
**Confidence:** 5

**Summary:**

The paper tackles the problem of imputation and generation in a single framework utilizing diffusion model with an additional transformer architecture. Experiments are shown on a widerange of datasets as well as competing models to highlights the benefits of the proposed approach.

**Strengths:**

Overall:
The paper is easy to read and the contribution is simple but effective. The experiments cover a wide range of datasets though not algorithms.

Pros:

(i) The paper extends TabDDPM to TabGenDDPM utilizing the transformer architecture which has been wildly succesful in other generative settings. The experiments confirm the benefits of the proposed approach. The additional benefit of covering both imputation and generation in the same framework enables a wide range of usecases in real-world settings.
(ii) Experiments cover around 10 datasets with varying number of rows and feature sizes and in almost all cases the proposed method is the best and sometimes by a big margin.

**Weaknesses:**

Cons:

(a) Some of the other competing methods like AIM, CTAB-GAN+ and others are not compared in the paper.
(b) The number of features in the datasets are few. HELOC has the highest with only 21 features and it is unclear how this framework performs when the feature set is large.

**Questions:**

(1) What is the running time of the proposed approach and how does it compare with the other state-of-the-art algorithms?
(2) How does it perform when the feature set is large and/or the number of samples is small?
(3) How does it work in augmentation tasks where the training is a mix of real + synthetic and testing on real?

---

> ### Author Response · Authors · 2023-11-22
>
> Dear Reviewer TceJ,
>
> Thank you so much for taking the time to review our submission. We will try to address all questions:
>
>
> **Q (1)**: What is the running time of the proposed approach and how does it compare with the other state-of-the-art algorithms?
>
> **A**: We have not conducted rigorous timing measurements, but the approximate order of runtime, from fastest to slowest, is as follows: TVAE < CTGAN < TabDDPM < TabGenDDPM. The difference in running time between TVAE and CTGAN is minimal, similar to that between TabDDPM and TabGenDDPM.
>
> **Q (2)**: How does it perform when the feature set is large and/or the number of samples is small?
>
> **A**: The dataset with the smallest sample size and the largest feature set that we have tested is HELOC. For this dataset our proposed method outperforms the baselines by a greater margin than for the other datasets.
>
> **Q (3)**: How does it work in augmentation tasks where the training is a mix of real + synthetic and testing on real?
>
> **A**: We conducted some preliminary tests on smaller datasets, where combining real and synthetic data generally improved test scores. However, these were not included in the paper to maintain consistency with other methods and the oracle baseline. While we haven't explored this in depth, it's an area we're considering for future research. In Table 2 of our results, we demonstrate that for the “Insurance” dataset, an XGBoost model trained on synthetic data outperforms one trained on real data. This suggests that synthetic data could indeed be beneficial for augmentation purposes.

---

### Meta-Review · Area_Chair_pzSN · 2023-12-12

**Metareview:**

The paper introduces a transformer-based diffusion mode, TabGenDDPM, for tabular data imputation and synthetic data generation. The proposed method extends TabDDPM by using a transformer architecture and a novel masking strategy, which allow the model to condition on a target variable and a subset of the features. In the empirical evaluation, the proposed method performs better than the baseline methods on various datasets. The reviews found the paper well-written, the extension of TabDDPM to have some novelty and be well-explained, and the empirical results to be strong. However, the reviewers also found the innovation to be incremental and the overall contribution of the paper to be limited, and raised concerns about scalability and applicability to more complex datasets, as the largest dataset tested had only 21 features, and raised concerns about the absence of rigorous running time measurements.

**Justification For Why Not Higher Score:**

While this is a borderline case, and the paper has several interesting and relevant contributions, it was not accepted due to its incremental innovation and limited overall contribution.

**Justification For Why Not Lower Score:**

N/A

---

### Decision · Program_Chairs · 2024-01-16

Reject